# In Vitro Antimicrobial Activity of Isopimarane-Type Diterpenoids

**DOI:** 10.3390/molecules25184250

**Published:** 2020-09-16

**Authors:** Vera M. S. Isca, Joana Andrade, Ana Sofia Fernandes, Paulo Paixão, Clara Uriel, Ana María Gómez, Noélia Duarte, Patrícia Rijo

**Affiliations:** 1CBIOS, Universidade Lusófona Research Center for Biosciences, Campo Grande 376, 1749-024 Lisboa, Portugal; vera.isca@ulusofona.pt (V.M.S.I.); joana.andrade@ulusofona.pt (J.A.); ana.fernandes@ulusofona.pt (A.S.F.); 2Research Institute for Medicines (iMed.ULisboa), Faculty of Pharmacy, Universidade de Lisboa, Av. Prof. Gama Pinto, 1649-003 Lisboa, Portugal; ppaixao@ff.ulisboa.pt (P.P.); mduarte@ff.ulisboa.pt (N.D.); 3Instituto de Química Orgánica, CSIC, Juan de la Cierva 3, E-28006 Madrid, Spain; clara.uriel@csic.es (C.U.); ana.gomez@csic.es (A.M.G.)

**Keywords:** isopimaranes, antimicrobial, MDA-MB-231 cells

## Abstract

The antimicrobial evaluation of twelve natural and hemisynthetic isopimarane diterpenes are reported. The compounds were evaluated against a panel of Gram-positive bacteria, including two methicillin-resistant *Staphylococcus aureus* (MRSA) strains and one vancomycin-resistant *Enterococcus* (VRE) strain. Only natural compounds 7,15-isopimaradien-19-ol (**1**) and 19-acetoxy-7,15-isopimaradien-3β-ol (**6**) showed promising results. Isopimarane (**1**) was the most active, showing MIC values between 6.76 µM against *S. aureus* (ATCC 43866) and 216.62 µM against *E. faecalis* (FFHB 427483) and *E. flavescens* (ATCC 49996). Compound (**6**) showed moderated activity against all tested microorganisms (MIC between value 22.54 and 45.07 µM). These compounds were found to be active against the methicillin-sensitive strains of *S. aureus* (CIP 106760 and FFHB 29593), showing MIC values of 13.55 (**1**) and 22.54 (**6**) µM. Both compounds were also active against vancomycin-resistant *E. faecalis* (ATCC 51299) (MIC values of 54.14 and 45.07 µM, respectively). In addition, the cytotoxicity of nine compounds 7,15-isopimaradien-3β,19-diol (**2**); mixture: 15-isopimarene-8β-isobutyryloxy-19-ol and 15-isopimarene-8β-butyryloxy-19-ol (**3**); 3β-acetoxy-7,15-isopimaradiene-19-ol (**5**); 19-acetoxy-7,15-isopimaradiene-3β-ol (**6**); 3β,19-diacetoxy-7,15-isopimaradiene (**8**); 15-isopimarene-8β,19-diol (**9**); 19-*O*-β-d-glucopyranoside-7,15-isopimaradiene (**10**); lagascatriol-16-*O*-β-d-glucopyranoside (**11**) and lagascatriol-16-*O*-α-d-mannopyranoside (**12**) was evaluated in the human breast cancer cell line MDA-MB-231. Isopimarane (**2**) was the only compound showing some cytotoxicity. The IC_50_ value of compound (**2**) was 15 µM, suggesting a mild antiproliferative activity against these breast cancer cells.

## 1. Introduction

Antibiotic resistance is nowadays an increasingly serious threat to global public health, leading to longer hospital stays, higher medical costs and increased mortality in all parts of the world [1]. A growing list of infections are becoming more difficult, and sometimes impossible, to treat. In particular, methicillin-resistant *Staphylococcus aureus* (MRSA) and vancomycin-resistant *Enterococcus* (VRE) are a worldwide problem in clinical medicine, being the former the leading cause of mortality due to bacterial infections [2]. Therefore, the research and development of new and effective antibacterial agents is considered a priority by all the world health agencies [3]. An important strategy to overcome this need is the continuous research on natural products, which have historically been of crucial importance in the identification of antibacterial drugs [4,5].

Pimaranes are tricycle diterpenes with different stereochemistry features and are biosynthetically related to labdane terpenoids. They have been found in several plants, namely in Lamiaceae and Asteraceae families [6,7]. Currently, pimarane-type diterpenoids are attracting significant scientific and therapeutic interest due to various biological activities reported, such as antitumor [8,9], anti-inflammatory and analgesic activity [10,11,12], and as a protein tyrosine phosphatase inhibitor [13]. In addition, the antibacterial activities of diverse pimaranes, mainly against antibiotic-resistant clinical strains, have been published [14,15,16,17,18].

In a previous study, we reported the isolation and structure characterization of isopimarane-type diterpenes from *Aeollanthus rydingianus* (Lamiaceae) as well as their antimicrobial activity against a set of Gram-positive and Gram-negative bacteria. Two compounds, namely 7,15-isopimaradien-19-ol (**1**, akhdarenol) and 19-acetoxy-7,15-isopimaradien-3β-ol (**6**) showed promising antimicrobial activity against *Staphylococcus aureus* and *Enterococcus hirae* [19]. Continuing our search for effective antimicrobial compounds, herein we report the preparation and structure elucidation of five new acyl and glycosyl derivatives of natural isopimaranes. These new derivatives and those previously isolated were assayed against an enlarged panel of bacteria, which included two MRSA strains. In addition, the cytotoxicity of the compounds was evaluated in the human breast cancer cell line MDA-MB-231.

## 2. Results and Discussion

In a previous study, a preliminary assay was conducted to assess the antimicrobial activity of a set of isopimarane diterpenes isolated from *Aeollanthus rydingianus*: 7,15-isopimaradien-19-ol (**1**), 7,15-isopimaradien-3β,19-diol (**2**), mixture of 3β-isobutyryloxy- and 3β-butyryloxy-8β-hydroxy-15-isopimarene (**3**), 3β-acetoxy-7,15-isopimaradiene (**4**), 3β-acetoxy-7,15-isopimaradien-19-ol (**5**), 19-acetoxy-7,15-isopimaradien-3β-ol (**6**) [19]. Compounds **1** and **6** were found to be very active against the methicillin-sensitive strain of *S. aureus* (ATCC 25923), showing MIC values of 3.90 and 15.6 µg mL^−1^, respectively. Both compounds were also active against *E. hirae* (MIC value of 7.81 µg mL^−1^).

In this work, and in order to study the effect of substituents on the antimicrobial activity of isopimarane-type diterpenes, compounds **1** (7,15-isopimaradien-19-ol) and **2** (7,15-isopimaradien-3β,19-diol) previously isolated [19] were derivatized to obtain the new acetylated compounds **7** (19-acetoxy-7,15-isopimaradiene) and **8** (3β,19-diacetoxy-7,15-isopimaradiene), and the glucosyl derivative **10** (19-glucosyl-7,15-isopimaradiene). Derivatives **11** (lagascatriol-16-*O*-β-d-glucopyranoside) and **12** (lagascatriol-16-*O*-α-d-mannopyranoside) were also obtained.

The structure elucidation of these new compounds was essentially based on comparison of NMR spectra with those of the corresponding parent alcohol. In this way, the ^1^H- and ^13^C-NMR spectra of compounds **7** and **8** revealed additional signals of acetyl ester residues (δ_H_ 0.86 s; δ_C_ 20.9 and 171.3 for compound **7**; δ_H_ 0.88 *s* and 1.01 s; δ_C_ 21.4, 21.6, 17. 8 and 171.1 for compound **8**). The presence of glucosyl and mannosyl residues of compounds **10**–**12** were also confirmed by NMR experiments, namely monodimensional ^1^H- and ^13^C-NMR spectra that evidenced the characteristic signals corresponding to the anomeric protons H-1′ (δ_H_ 4.21–4.77, *d*) and anomeric carbons C-1′ (δ_C_ 105.4 and 104.3 glucosyl derivatives **10** and **11**, and δ_C_ 102.4 for mannosyl derivative **12**). All the assignments were in agreement with COSY, HSQC and HMBC spectra. The connection of the sugar residues to the C-16 position of lagascatriol was further evidenced by the HMBC spectrum, which showed correlations between the C-1′ carbon (δ_C_ 102.4) and both C-16 protons (at δ_H_ 3.92 and 3.32), and between the H-1′ proton (δ_H_ 4.77) and the C-16 carbon (at δ_H_ 69.8).

Acetyl derivatives were synthesized aiming to clarify the role of the 3- and 19-oxygen atoms (hydroxyl versus acetoxyl groups) as either hydrogen donor or acceptor atom (**1**/**7**, **2**/**8**), on the antimicrobial activity. The effect of a high hydrophilicity, achieved through the introduction of *O*-monoglycosyl groups (**10**, **11**, **12**), was also evaluated.

All compounds were tested against an enlarged panel of Gram-positive bacteria (*S. aureus, S. epidermis, E. faecalis*, *E. faecium*, *E. flavescens* and *E. hirae.* Several strains of *S. aureus* and *E. faecalis* were assayed, including two methicillin-resistant *S. aureus* (CIP 106,760 and FFHB 29593) and one vancomycin-resistant *E. fecalis* (ATCC 51299). A previous antimicrobial screening (Rijo et al. 2009) of compounds **1** and **6**, against Gram-negative bacteria, *Mycobacterium smegmatis* and *Candida albicans* indicated that they were inactive.

Isopimaradienol **1** was the most active, showing MIC values between 6.76 µM (1.95 µg mL^−1^) against *S. aureus* (ATCC 43866) and 216.62 µM (62.50 µg mL^−1^) against *E. faecalis* (FFHB 427483) and *E. flavescens* (ATCC 49996) (Table 1). In general terms, compound **1** showed high activity against all *S. aureus* tested; additionally, it showed a potent growth inhibitory activity against both the methicillin-sensitive and -resistant *S. aureus* strains and vancomycin-resistant *E. faecalis* with MIC values of 6.76, 27.07 and 54.14 µM (1.95, 7.81 and 15.62 µg mL^−1^), respectively. This isopimarane bears a C-19 hydroxyl group to which the activity was attributed in the related *ent*-isopimara-9(11),15-diene-19-ol [19]. Conversely, its 19-*O*-acetyl ester **7** and its 19-*O*-glucosyde **10** were inactive against all the bacteria tested, except for *S. aureus* ATCC 43,866 strain, which compound **7** had an MIC of 11.83 (3.91 µg mL^−1^) (Table 1). In the same way, 3,19-diol **2** only was bioactive against *S. aureus* ATCC 43,866 and ATCC 6538 (MICs 51.30 µM). Despite this, natural 19-acetyl-3-ol **6** that has a C-3 hydroxyl group and a C-19 acetoxyl substituent instead a C-19 hydroxyl group showed high MIC values (22.54 to 45.07 µM) against all the Gram-positive bacteria assayed and an effective antibacterial active against the methicillin-sensitive strains of *S. aureus* (MICs of 22.54 µM) and vancomycin-resistant *E. faecalis* (MIC 45.07 µM). No other combination of the polar hydroxyl and acetoxyl groups on C-3 and C-19 resulted in significative bioactivity. When the acetylation occurs only in the 3-*O* position, as in the analogous **5**, or in both 3- and 19-hydroxyl groups, as in compound **8,** the activity significantly drops (MICs 360.69 and 321.67 µM, respectively). Both 8-hydroxyisopimaranes **3** and **9**, and the glycosyl derivatives of the lagascatriol **11** and **12**, did not reveal any activity in the tests carried out (Table 1).

### 2.1. Cytotoxicity Evaluation

The cytotoxicity of the compounds **1** to **3**, **5**, **6**, **8** to **12** was evaluated in MDA-MB-231 cells, a human cell line routinely used as a model for aggressive breast cancer. For compounds **4** and **7**, only little amounts were available, precluding their inclusion in the cytotoxicity assays. The compounds were initially screened at a final concentration of 10 μM. The compounds **1**, **3**, **5**, **6**, **8**–**12** were not cytotoxic to MDA-MB-231 cells under the conditions tested. Such results show that these compounds are devoid of considerable anticancer activity against MDA-MB-231 cells. However, for a complete screening of their anticancer properties, cytotoxic effects should be studied in additional cell lines, representative of other cancer types.

As 7,15-isopimaradien-19-ol **1** was the only compound showing some cytotoxicity at 10 μM, the concentration-response profile of this compound was evaluated (data not shown). The IC_50_ value of compound **1** was 15 µM, suggesting a mild antiproliferative activity against these breast cancer cells. Interestingly, this compound has shown an IC_50_ value of 56 μM in normal-like cells MRC-5 (data not shown), indicating some selectivity for cancer cells.

### 2.2. Structure-Activity Modeling

Although the presence of an -OH group at the 3′ or 4′ position of the molecule seems to be essential for the existence of antimicrobial activity, the reduced number of compounds (*n* = 7) with MIC values showing activity against *S. aureus* (ATCC 25923) limits the identification of relevant molecular descriptors. As can be seen (Table 2), only a short number of uncorrelated molecular descriptors were able to be used for the analysis, and of these, only the Number of Rotatable Bonds (RBN) presents a statistical significant positive correlation (*p* = 0.049) with the Log(MIC) values. Since the low aqueous solubility appears to be responsible for the reduction of oral bioavailability of the different compounds, a molecular optimization in this respect should take this into consideration.

## 3. Materials and Methods

### 3.1. General Experimental Procedures

All solvents were dried according to published methods and distilled prior to use. All the other reagents were obtained from commercial suppliers and were used without further purification. Flash column chromatography (CC) was performed on silica gel (Merck 9385). Merck silica gel 60 F254 plates were used in analytical TLC, with visualization under UV light (λ = 254 and 366 nm) and by spraying with H_2_SO_4_/MeOH (1:1), followed by heating. For preparative TLC chromatography, 20 × 20 cm × 0.5 mm silica plates were used (Merck 1.05774). NMR spectra were recorded on a Brüker ARX-400 NMR spectrometer (^1^H 400 MHz; ^13^C 100.61 MHz), or on a Varian System NMR spectrometer (^1^H 500 MHz; ^13^C 125 MHz) using CDCl_3_ or CD_3_OD, as solvents. All tested compounds were purified to ≥95% purity as determined by HPLC (HPLC-DAD Agilent Technologies 1200 Infinity Series system with the ChemStation software in combination with a LiChrospher^®^ 100, RP-18 (5 m) Merck column. The volume of injection of the samples was of 20 μL and a gradient composed of solution A (methanol), solution B (acetonitrile) and solution C (0.3% trichloroacetic acid in water), as follows, was used to carry out the analysis: 0 min, 15% A, 5% B and 80% C; 20 min, 70% A, 30% B and 0% C; 25 min, 70% A, 30% B and 0% C; and 28 min, 15% A, 5% B and 80% C. The flow rate was set at 1 mL min^−1^. The time of analysis was of 33 min, including the stabilization of the RP-18 column. The quantification for the all samples was performed in triplicate.

### 3.2. Compounds Tested

Twelve diterpenes with isopimarane skeleton, whose structures are presented in Figure 1, were tested. Compounds **1**–**6** and **9** were isolated from *Aeollanthus rydingianus* aerial parts [19]. The benzoylated glycosides of compounds **10**–**12** were prepared as described by Uriel et al. [20]. Compounds **7**, **8** and **10**–**12** were prepared as described below. All compounds were solubilized in DMSO and further tested for their biological assays.

#### Preparation of Isopimarane Derivatives **7** and **8**

A solution of parental alcohols **1** (1 eq.) or **2** (2 eq.) in dry pyridine and acetic anhydride was stirred at room temperature until the reaction was complete. The reaction mixture was then diluted with CH_2_Cl_2_ and successively washed with water and NaOH solution (5%), and the organic layer dried over anhydrous Na_2_SO_4_. The residue was purified by preparative TLC (CH_2_Cl_2_:CH_3_COCH_3_, 98:2) to afford compounds **7** and **8**, respectively. After purification, the compounds were extracted from silica gel with CH_2_Cl_2._

*19-acetoxy-7,15-isopimaradiene* (**7**), ^1^H-NMR (400 MHz, CDCl_3_): δ 5.78 (1H, dd, *J* = 6.8, 14.2 Hz, H-15), 5.35 (1H, dd, *J* = 1.2, 1.6 Hz, H-7), 4.94 (1H, d, *J* = 1.2 Hz, H-16a), 4.88 (1H, td, *J* = 1.3, 10.3 Hz, H-16b), 4.36 (1H, d, *J* = 10.8 Hz, H-19a), 3.98 (1H, m, H-19b), 2.05 (3H, s, 19-OCOCH_3_), 1.99 (2H, m, H-6a,b), 1.94 (1H, m, H-14a), 1.91 (1H, m, H-14b), 1.77 (1H, m, H-12a), 1.67 (1H, m, H-9), 1.55 (1H, m, H-11a), 1.46 (1H, m, H-1β), 1.44 (3H, m, H-2β), 1.33 (1H, m, H-11b), 1.30 (1H, m, H-5), 1.02 (1H, m, H-3), 1.00 (1H, m, H-2α), 0.96 (3H, s, 18-CH_3_), 0.86 (3H, s, 17-CH_3_), 0.86 (3H, s, 19-OCOCH_3_). ^13^C-NMR (100 MHz, CDCl_3_) δ171,3 (19-OCOCH_3_), 150.2 (C-15), 135.5 (C-8), 121.2 (C-7), 109,1 (C-16), 66.8 (C-19), 51.9 (C-9), 51.2 (C-5), 45.9 (C-14), 39.5 (C-3), 36.7 (C-13), 36.7 (C-13), 36.2 (C-4), 36 (C-1), 35.9 (C-12), 35.2 (C-10), 27.2 (C-18), 22.9 (C-6), 21.3 (C-17), 20.9 (19-OCOCH_3_), 20.2 (C-11), 18.2 (C-2), 16.0 (C-20).

*3*β,*19-diacetoxy-7,15-isopimaradiene* (**8**), ^1^H-NMR (400 MHz, CDCl_3_): δ 5.79 (1H, dd, *J* = 10.7, 17.5 Hz, H-15), 5.35 (1H, d, *J* = 1.8 Hz, H-7), 4.94 (1H, d, *J* = 1.3 Hz, H-16a), 4.88 (1H, td, *J* = 1.3, 9.5 Hz, H-16b), 4.59 (1H, m, H-3α), 4.47 (1H, d, *J* = 11.7 Hz, H-19a), 4.24 (1H, d, *J* = 11.7 Hz, H-19b), 2.06 (3H, s, 3β-OCOCH_3_), 2.04 (3H, s, 19-OCOCH_3_), 2.04 (2H, m, H-6a,b), 1.94 (1H, m, H-14b), 1.92 (1H, m, H-14a), 1.90 (1H, m, H-1β), 1.68 (2H, m, H-2a,b), 1.66 (1H, m, H-9), 1.54 (1H, m, H-11), 1.47 (1H, m, H-12a), 1.38 (1H, m, H-5), 1.35 (2H, m, H-12b, 11b), 1.28 (1H, m, H-1α), 1.01 (3H, m, 18-CH_3_), 0.88 (3H, s, 20-CH_3_), 0.85 (3H, s, 17-CH_3_). ^13^C-NMR (100 MHz, CDCl_3_) δ 171,1 (3-OCOCH_3_), 170.8 (19-OCOCH_3_), 150.3 (C-15), 135.6 (C-8), 121.2 (C-7), 109.5 (C-16), 80.5 (C-3), 64.8 (C-19), 51.9 (C-9), 51.2 (C-5), 45.9 (C-14), 40,8 (C-4), 37.7 (C-1), 36.9 (C-13), 36.1 (C-12), 35.2 (C-10), 24.2 (C-2), 23.7 (C-6), 22.8 (C-18), 21.6 (3-OCOCH_3_), 21.4 (19-OCOCH_3_), 21.3 (C-17), 20.5 (C-11), 15.6 (C-20).

### 3.3. General Procedure for Debenzoylation

The benzoylated glycoside was dissolved in MeOH:THF (1:1) (20 mL/mmol), a 0.1 M solution of MeO^−^Na^+^ in MeOH was added until pH 8, the reaction was stirred at room temperature and, when TLC showed the disappearance of the starting material, resine H^+^ (IR-120) was added until pH = 7. The mixture was filtered and concentrated. The residue was purified by flash silica gel column chromatography.

*19-O-*β-*d-glucopyranosyl-7,15-isopimaradiene* (**10**), ^1^H-NMR (500 MHz, CD_3_OD): δ 5.80 (1H, dd, *J* = 10.8, 17.5 Hz, 15-H), 5.36 (1H, m, H-7), 4.92 (1H, dd, *J* = 1.5, 17.6 Hz; H-16a), 4.85 (1H, dd, *J* = 1.4, 10.8 Hz, H-16b), 4.21 (1H, d, *J* = 7.8 Hz; H-1′), 3.88 (2H, m, H-6′a, H-19a), 3.73 (1H, dd, *J* = 9.4, 0.5 Hz; H-19b), 3.68 (1H, dd, *J* = 5.3, 11.9 Hz, H-6′b), 3.34–3.25 (3H, m, H-3′, H-4′, H-5′), 3.17 (1H, dd, *J* = 7.7, 9.0 Hz, H-2′), 2.09 (1H, m, H-3a), 1.99 (2H, m, H-6a, H-6b), 1.97 (1H, m, H-14a),1.88 (1H, m, H-14b), 1.87 (1H, m, H-1a), 1.68 (1H, m, H-9a), 1.57 (1H, m, H-11a), 1.55 (1H, m, H-2a), 1.47 (1H, m, H-12a), 1.42 (1H, m, H-2b), 1.38 (1H, m, H-12b), 1.37 (1H, m, H-11b), 1.26 (1H, m, H-5a), 1.07 (1H, m, H-1b), 1.00 (3H, s, 18-CH_3_), 0.92 (1H, m, H-3b), 0.89 (3H, s, 20-CH_3_), 0.86 (3H, s, 17-CH_3_). ^13^C-NMR (125 MHz, CD_3_OD): 150.1 (C-15), 136.6 (C-8), 122.9 (C-7), 109.7 (C-16), 105.4 (C-1′), 78.2, 77.8 (C-3′,C-4′), 75.3 (C-2′), 73.6 (C-19), 71.7 (C-5′), 62.8 (C-6′), 53.7 (C-9), 53.1 (C-5), 47.3 (C-14), 41.1 (C-1), 38.4 (C-4 or C-10), 37.9 (C-10 or C-4), 37.4 (C-13), 36.8 (C-3), 36.5 (C-12), 28.3 (C-18), 24.2 (C-6), 22.0 (C-17), 21.5 (C-11), 19.5 (C-2), 16.7 (C-20).

*Lagascatriol-16-O-*β-*d-glucopyranoside* (**11**), ^1^H-NMR (500 MHz, CD_3_OD): δ 5.50 (1H, br d, *J* = 5.7 Hz, H-6), 4.26 (1H, d, *J* = 7.9 Hz; H-1′), 3.87 (1H, br d, *J* = 11.9 Hz, H-6′a), 3.80 (1H, m, H-16), 3.66 (1H, ddd, *J* = 11.9, 4.1, 1.4 Hz; H-6′b) 3.58 (1H, dd, *J* = 11.7, 4.6 Hz; H-11β), 3.36 (2H, m, H-3′, H-15), 3.27 (3H, m, H-4′, H-5′, H-16b), 3.21 (1H, dd, *J* = 7.9, 9.2 Hz; H-2′), 2.55 (1H, br d, *J* = 13.9 Hz, H-1β), 2.18 (1H, br d, *J* = 12.3 Hz; H-10β), 1.78 (1H, m; H-7α), 1.67 (1H, m; H-7β), 1.63 (1H, qt, *J* = 13.3 Hz, 3.3 Hz, H-2β), 1.20 (1H, td, *J* = 13.2, 4.4 Hz; H-3α), 1.05 (3H, s, 18-CH_3_), 1.04 (3H, s, 19-CH_3_), 0.98 (3H, s; 17-CH_3_), 0.70 (3H, s, 20-CH_3_), 1.01, 1.53, 1.40, 0.99, 1.32, 1.53 and 1.42 (H-1_α_, H-2_α_, H-3_β_, H-8_β_, H-12, H-14a and H-14b). ^13^C-NMR (125 MHz, CD_3_OD): δ 148.1 (C-5), 116.9 (C-6), 104.3 (C-1′), 79.8 (C-15), 78.8 (C-11), 78.0 (C-3′), 77.9 (C-5′), 75.1 (C-2′), 71.7 (C-16), 71.6 (C-4′), 62.7 (C-6′), 48.2 (C-10), 42.3 (C-3), 41.5 (C-9), 39.9 (C-14), 38.8 (C-13), 37.4 (C-8), 37.3 (C-4), 37.1 (C-12), 32.1 (C-1), 30.6 (C-18 and C-7), 29.0 (C-19), 23.5 (C-2), 20.4 (C-17), 6.6 (C-20).

*Lagascatriol-16-O-*α-*d-mannopyranoside* (**12**), ^1^H-NMR (500 MHz, CD_3_OD): δ 5.49 (1H, dt, *J* = 6.4, 1.7 Hz; H-6), 4.77 (1H, d, *J* = 1.7 Hz, H-1′), 3.92 (1H, m, H-16a), 3.86 (1H, dd, *J* = 1.7, 3.4 Hz, H-2′), 3.83 (1H, dd, *J* = 11.8, 2.2 Hz; H-6a’), 3.71 (1H, dd, *J* = 3.4, 9.2 Hz; H-3′), 3.69 (1H, dd, *J* = 11.8, 6.1 Hz; H-6′b), 3.58 (1H, t, *J* = 9.1 Hz; H-4′), 3.58 (1H, dd, *J* = 12.0, 4.1 Hz; H-11β), 3.53 (1H, ddd, *J* = 9.1, 2.2, 6.1 Hz, H-5′), 3.32 (2H, m, H-16b and H-15), 2.54 (1H, dddd, *J* = 13.0, 3.4 Hz, H-1β), 2.18 (1H, br ddd, *J* = 12.7, 3.4, 1.7 Hz, H-10β), 1.78 (1H, dddd, *J* = 14.2, 11.0, 1.7, 3.4 Hz, H-7α), 1.67 (1H, m, H-7β), 1.63 (1H, qt, *J* = 13.2, 3.4 Hz, H-2β), 1.53 (2H, m; H-2α and H-14a), 1.42 (1H, m; H-14b), 1.41 (1H, m, H-3β), 1.30 (2H, m, H-12), 1.20 (1H, td, J = 13.2, 4.1 Hz, H-3α), 1.05 (3H, s; 18-CH_3_), 1.03 (3H, s, 19-CH_3_), 1.00 (1H, m, H-1α), 0.98 (1H, m, H-8β), 0.97 (3H, s, 17-CH_3_), 0.69 (3H, s, 20-CH_3_). ^13^C-NMR (125 MHz, CD_3_OD): δ 148.1 (C-5), 116.9 (C-6), 102.4 (C-1′), 80.4 (C-15), 78.8 (C-11), 74.8 (C-5′), 72.6 (C-3′), 72.0 (C-2′), 69.8 (C-16), 68.7 (C-4′), 63.0 (C-6′), 48.2 (C-10), 42.3 (C-3), 41.5 (C-9), 39.9 (C-14), 39.6 (C-13), 37.4 (C-8), 37.3 (C-4), 37.1 (C-12), 32.1 (C-1), 30.6 (C-18), 29.6 (C-7), 29.0 (C-19), 23.5 (C-2), 20.3 (C-17), 6.6 (C-20).

### 3.4. Antimicrobial Assays

#### 3.4.1. Microbial Strains

*Staphylococcus aureus* ATCC 25923, *S. aureus* ATCC 43866, *S. aureus* ATCC 700699, *S. aureus* CIP 106,760 (MRSA—methicillin-resistant *S. aureus*), *S. aureus* FFHB 29,593 (MRSA), *Enterococcus faecalis* ATCC 51,299 (low level VRE—vancomycin-resistant *Enterococcus*), *E. faecalis* FFHB 427483, *E. faecalis* CIP 106996, *E. faecium* FFHB 435628, *E. flavescens* ATCC 49996, *E. hirae* CIP 5855. FFHB species were identified and deposited on the Microbiology Laboratory of the Faculty of Pharmacy, Lisbon University, from clinical isolates of Hospital do Barreiro.

#### 3.4.2. Microdilution Method

The MIC values of compounds, against the tested strains, were performed by means of the twofold serial broth microdilution assay [21]. The compounds, dissolved in DMSO, were diluted at concentrations ranging from 500 to 0.49 µg/mL with a Müeller-Hinton broth medium for bacteria, and a Sabouraud broth medium for the yeast strain. The antimicrobial activity of the solvent was evaluated, and control antibiotics were included. The MIC value was taken as the lowest concentration of compound that inhibited the growth of the test organisms after 24 h of incubation at 37 °C. The bacterial growth was measured with an absorbance microplate reader set to 630 nm (ELX808TM-BioteK, Winooski, VT, USA). Assays were carried out in triplicate for each tested microorganism.

### 3.5. Cytotoxicity Assays

#### 3.5.1. Cell Culture

MDA-MB-231 cells were cultured in DMEM (Sigma Aldrich, St. Louis, MO, USA) supplemented with 10% fetal bovine serum, 100 U/mL penicillin, and 0.1 mg/mL streptomycin [22]. The cultures were maintained at 37 °C, under a humidified atmosphere containing 5% CO_2_.

#### 3.5.2. Crystal Violet (CV) Staining Assay

Cell viability was evaluated by the crystal violet (CV) staining assay. Approximately 6 × 10^3^ cells in 190 μL of culture medium per well, were plated in 96-well plates, and allowed to grow for 24 h. Cells were then exposed to the different compounds for 48 h. The CV assay was carried out according to a previously described protocol [23]. DMSO 5% (*v*/*v*) was used as positive control. At least two independent experiments were performed, each comprising four replicate cultures.

### 3.6. Structure-Activity Modelling

A relationship between some molecular descriptors, namely constitutional, topological, geometrical, WHIM, functional group counts and molecular properties descriptors, was evaluated based on the activity of the compounds against *Staphylococcus aureus* (ATCC 25923). MIC values were considered with units of µM and a logarithmic transformation was performed. Only compounds with activities <125 µg mL^−1^ were considered and linear relationships between the molecular descriptors and the Log (MIC) were calculated. Only non-correlated molecular descriptors (*p*-values < 0.05) were considered in the analysis.

## 4. Conclusions

Only the pimaranes **1** and **6** showed promising antibacterial activity, despite the efforts to find more potent antimicrobial analogous. The antimicrobial activity against *S. aureus* and *E. hirae* was already known [19]; however, this time, the range of Gram-positive bacteria was higher. Within this, compound **1** showed the most potent antibacterial activity of the set of tested compounds. Isopimarane **6** presented high antimicrobial activity against the Gram-positive assayed. Among the acetylated and glicosyl derivates, only 19-*O*-acetyl ester **7** and 3,19-diol **2** showed some antibacterial activity. These results indicate that the additional substituents incorporated did not lead to an improved antimicrobial activity, probably because the acetyl group drastically changes the donor/acceptor abilities and glycosyl substituents diminished the lipophilic properties of these molecules. Further studies are being carried out to improve the bioactivity. Isopimarane **1** showed some cytotoxicity. The IC_50_ value of compound **1** was 15 µM, suggesting a mild antiproliferative activity against these breast cancer cells. The compounds **1**, **3**, **5**, **6**, **8**–**12** were not cytotoxic to MDA-MB-231 cells under the conditions tested; however, for a complete screening of their cytotoxic effects, they should be studied in additional cell lines, representative of other cancer types.

## Figures and Tables

**Figure 1 molecules-25-04250-f001:**
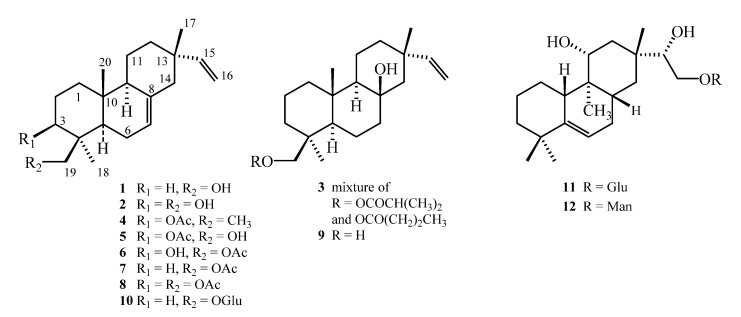
Structures of compounds: (**1**) 7,15-isopimaradien-19-ol (akhdarenol); (**2**) 7,15-isopimaradien-3β,19-diol (virescenol B); (**3**) mixture: 15-isopimarene-8β-isobutyryloxy-19-ol and 15-isopimarene-8β-butyryloxy-19-ol; (**4**) 3β-acetoxy-7,15-isopimaradiene; (**5**) 3β-acetoxy-7,15-isopimaradiene-19-ol; (**6**) 19-acetoxy-7,15-isopimaradiene-3β-ol; (**7**) 19-acetoxy-7,15-isopimaradiene; (**8**) 3β,19-diacetoxy-7,15-isopimaradiene; (**9**) 15-isopimarene-8β,19-diol; (**10**) 19-*O*-β-d-glucopyranoside-7,15-isopimaradiene; (**11**) lagascatriol-16-*O*-β-d-glucopyranoside; (**12**) lagascatriol-16-*O*-α-d-mannopyranoside.

**Table 1 molecules-25-04250-t001:** MIC (μM) values of isopimaranes **1**–**12** against *Staphylococcus aureus* and *Enterococcus* strains.

	*S. aureus*						*E. faecalis*			*E. faecium*	*E. flavescens*	*E. hirae*
	ATCC 25923	ATCC 43866	ATCC 700699	CIP 106760	FFHB 29593	ATCC 6538	ATCC 51299	CIP 106996	FFHB 427483	FFHB 435628	ATCC 49996	CIP 5855
**1**	13.55	6.76	27.07	13.55	13.55	13.55	54.14	27.07	216.62	>433.25	216.62	27.07
**2**	410.55	51.30	>821.10	410.55	410.55	51.30	>410.55	>410.55	nt	>410.55	>410.55	>410.55
**3**	>676.71	>676.71	>676.71	>676.71	>676.71	nt	>338.36	nt	>338.36	>338.36	>338.36	>338.36
**4**	>725.61	181.40	>725.61	>725.61	>725.61	nt	181.40	nt	181.40	>362.80	181.40	>362.80
**5**	>721.38	>721.38	>721.38	>721.38	>721.38	nt	>360.69	nt	180.34	>360.69	>360.69	>360.69
**6**	45.07	22.54	45.07	22.54	22.54	nt	45.07	nt	22.54	45.07	45.07	45.07
**7**	378.15	11.83	>756.29	378.15	378.15	189.07	>378.15	>378.15	>378.15	>378.15	nt	>378.15
**8**	321.67	321.67	>643.34	321.67	321.67	160.83	>321.67	>321.67	>321.67	>321.67	nt	>321.67
**9**	>815.69	>815.69	>815.69	>815.69	>815.69	nt	>407.84	nt	>407.84	>407.84	>407.84	>407.84
**10**	138.68	277.36	>554.72	277.36	277.36	138.68	nt	>277.36	nt	nt	nt	nt
**11**	257.89	257.89	257.89	257.89	257.89	128.95	>257.89	>257.89	>257.89	128.95	nt	>257.89
**12**	257.89	257.89	>515.78	257.89	257.89	128.95	>257.89	>257.89	>257.89	>257.89	nt	>257.89
Vancomycin	1.35	2.70	5.39	2.70	2.70	-	nt	21.56	1.35	0.68	2.70	0.68
Tetracycline	<1.10	281.26	140.63	70.31	<1.10	-	-	<1.10	70.31	<1.10	<1.10	<1.10
Ampicillin	<1.40	>715.51	<1.40	>715.51	357.76	-	-	<1.40	<1.40	>357.76	<1.40	<1.40
Methicillin	2.58	5.13	41.06	>657.17	>657.17	-	-	-	-	-	-	-
DMSO	3199.80	3199.80	3199.80	3199.80	3199.80	-	-	1599.90	1599.90	1599.90	1599.90	1599.90

nt—not tested.

**Table 2 molecules-25-04250-t002:** Correlation analysis between selected molecular descriptors and the MIC values against *Staphylococcus aureus* (ATCC 25923).

Compound	MW	nBM	RBN	MAXDP	ASP	Gu	nCp	LogMIC
**1**	288.52	2	2.0	3.987	0.502	0.172	4	1.131
**6**	346.56	3	4.0	4.786	0.489	0.185	4	1.654
**7**	330.56	3	4.0	4.358	0.491	0.164	4	2.578
**8**	388.60	4	6.0	4.759	0.461	0.212	5	2.507
**10**	450.68	2	5.0	4.385	0.499	0.181	4	2.142
**11/12**	484.70	1	5.0	5.420	0.708	0.179	5	2.411
r	0.560	0.208	0.813	0.506	0.201	0.236	0.528	
*p*-value	0.248	0.692	0.049	0.306	0.703	0.653	0.281	

MW—molecular weight; nBM—number of multiple bonds; RBN—number of rotatable bonds; MAXDP—maximal electrotopological positive. variation; ASP—asphericity; Gu—unweighted G total symmetry index; nCp—number of total primary C(sp^3^); r—correlation coefficient.

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
