# Peer review of "In Vitro Antimicrobial Activity of Isopimarane-Type Diterpenoids"

_molecules, 2020, doi:10.3390/molecules25184250_

Round 1

Reviewer 1 Report

The manuscript "In vitro antimicrobial activity of isopimarane-type diterpenoids" deals with the preparation and structure elucidation of five new acyl and glycosyl derivatives of natural isopimaranes. These new derivatives, together with other isopimarane-type diterpenes previously isolated from Aeollanthus rydingianus  were assayed against a panel of Gram-positive and Gram- negative bacteria.

The study addresses a topic of great interest which falls within the Journal’s scope. The manuscript is well organized and well conducted. However, the following issues need to be addressed:

1) Line 20-23: The Author should avoid to mention in the abstract compounds 2, 3, 5, and 8 to 12 because these molecules can not be known to the reader if not reading the entire article

2) Line 65, 66: The operating conditions of HPLC (or a reference) should be reported,

3) Figure 1: The Author should report the solubility of the tested compounds, 

4) Line 83, 84: The Author should report the solvent, or solvent mixture, used to extract compound 7 and 8 from the silica gel after purification by preparative TLC

Moreover, some typos, (e.g. line 163 "absorbance" instead of "absorvance", should be addressed.

Author Response

RESPONSE TO REVIEWERS 1 COMMENTS

Manuscript ID: molecules-928417

Title: In vitro antimicrobial activity of isopimarane-type diterpenoids

Authors: Vera Isca, Joana Andrade, Ana Sofia Fernandes, Paulo Paixão, Clara Uriel, Ana María Gómez, Noelia Duarte, Patrícia Rijo

Comments to the Authors

The manuscript "In vitro antimicrobial activity of isopimarane-type diterpenoids" deals with the preparation and structure elucidation of five new acyl and glycosyl derivatives of natural isopimaranes. These new derivatives, together with other isopimarane-type diterpenes previously isolated from Aeollanthus rydingianus were assayed against a panel of Gram-positive and Gram- negative bacteria.

The study addresses a topic of great interest which falls within the Journal’s scope. The manuscript is well organized and well conducted. However, the following issues need to be addressed:

Comment 1: Line 20-23: The Author should avoid to mention in the abstract compounds 2, 3, 5, and 8 to 12 because these molecules can not be known to the reader if not reading the entire article

Authors: We thank the reviewer for the suggestion. We change the reference to the compounds.

Comment 2: Line 65, 66: The operating conditions of HPLC (or a reference) should be reported,

Authors: We thank the reviewer for the comment. We include the operating conditions of HPLC.

Comment 3:  Figure 1: The Author should report the solubility of the tested compounds,

Authors: We thank the reviewer for the comment. We include this information in the manuscript.

Comment 4:  Line 83, 84: The Author should report the solvent, or solvent mixture, used to extract compound 7 and 8 from the silica gel after purification by preparative TLC

Authors: We thank the reviewer for the comment. We include this information in the manuscript.

Comment 5: Moreover, some typos, (e.g. line 163 "absorbance" instead of "absorvance", should be addressed.

Authors: We thank the reviewer for the comment. The word was changed accordingly.

Reviewer 2 Report

This manuscript describes the antimicrobial evaluation of 12 isopimarane diterpenes. Also, cytotoxicity of compounds 1 to 3, 5, 6, 8 to 12 was evaluated in the human breast cancer cell line. The manuscript is robust and fit within the scopes of the journal. Therefore, this referee believes that it deserves to be published in Molecules, pending the following minor points:

  • Mass spectrometry characterization for the final compounds is missing.
  • In lines 85 and 112 (nomenclature) there are typing errors
  • The cytotoxic effect of the most active compounds should be assayed in non tumoral mammalian cell lines (e.g. VERO or MDCK), in order to compare the selective toxicity against bacteria versus normal cells
  • Authors should report the MIC of isopimarenes 1 12 using micromolar concentrations [μM] instead of micrograms per mL [μg/mL], in order to compare relative potencies with the antibiotics used as positive controls

Author Response

RESPONSE TO REVIEWER 2 COMMENTS

Manuscript ID: molecules-928417

Title: In vitro antimicrobial activity of isopimarane-type diterpenoids

Vera Isca, Joana Andrade, Ana Sofia Fernandes, Paulo Paixão, Clara Uriel, Ana María Gómez, Noelia Duarte, Patrícia Rijo

Comments to the Authors

This manuscript describes the antimicrobial evaluation of 12 isopimarane diterpenes. Also, cytotoxicity of compounds 1 to 3, 5, 6, 8 to 12 was evaluated in the human breast cancer cell line. The manuscript is robust and fit within the scopes of the journal. Therefore, this referee believes that it deserves to be published in Molecules, pending the following minor points:

Comment 1: Mass spectrometry characterization for the final compounds is missing.

Authors: We thank the reviewer for the comment. This work is about twelve isopimarane compounds their antimicrobial activity and structure-activity modeling. The compounds 1-6 and 9 were isolated from Aeollanthus rydingianus aerial parts, which were used as starting material and are totally characterized in Rijo et al., 2009. Another issue is that the compounds 7,8 and 10-12 which are the only ones that have no mass spectrometry characterization were obtained from the natural ones and were obtained in very low quantities of products.

Comment 2: In lines 85 and 112 (nomenclature) there are typing errors

Authors: We thank the reviewer for the comment. The nomenclature was corrected.

Comment 3: The cytotoxic effect of the most active compounds should be assayed in non tumoral mammalian cell lines (e.g. VERO or MDCK), in order to compare the selective toxicity against bacteria versus normal cells.

Authors: We thank the reviewer for the comment. We have evaluated the cytotoxicity of compound 1 in MRC-5 cells, which are human lung normal-like fibroblasts. The obtained IC50 value was 56 μM, indicating higher toxicity to cancer cells than to the non-cancer ones. This aspect is now mentioned in the revised manuscript. Besides the evaluation of the cytotoxicity to cancer models, we agree that the study of the cytotoxicity in non-cancer cells would be informative to explore whether the compounds have antibacterial effects in concentrations that are non-toxic to human cells and can be thus further explored as potential antibiotics. However, to properly explore the safety of the compounds, the isolation and/or derivatization of sufficient amounts of the compounds is needed. Furthermore, not only cytotoxicity but also genotoxicity and other safety endpoints should be explored. A more in-depth study focused on the safety evaluation of the compounds is thus planned as future work. For all these results we decided to include the co-author Nuno Saraiva in the manuscript.

Comment 4: Authors should report the MIC of isopimarenes 1 – 12 using micromolar concentrations [μM] instead of micrograms per mL [μg/mL], in order to compare relative potencies with the antibiotics used as positive controls

Authors: We thank the reviewer for the comment. We include the new values in Table 1 of the manuscript.